# Untangling the tingle: Investigating the association between the Autonomous Sensory Meridian Response (ASMR), neuroticism, and trait & state anxiety

**Charlotte M. Eid** [ID] ◉, **Colin Hamilton** [ID] ‡, **Joanna M. H. Greer** [ID] ‡*◉

Department of Psychology, Northumbria University, Newcastle-upon-Tyne, United Kingdom

◉ These authors contributed equally to this work.
‡ CH and JMHG also contributed equally to this work.
* joanna.greer@northumbria.ac.uk

**Data Availability Statement:** All data files area available from the Northumbria University repository: https://doi.org/10.25398/rd.northumbria.16654846.

## Abstract

The Autonomous Sensory Meridian Response (ASMR) is an intensely pleasant tingling sensation originating in the scalp and neck and is elicited by a range of online video-induced triggers. Many individuals now regularly watch ASMR videos to relax, and alleviate symptoms of stress and insomnia, all which are indicative of elevated levels of anxiety. Emerging literature suggests that ASMR-capable individuals are characterised by high trait neuroticism, which is associated with a tendency to experience negative emotional states such as anxiety. To date however no literature has empirically linked these personality constructs and watching ASMR videos on the effect of reducing anxiety. In the current study, 36 ASMR-experiencers and 28 non-experiencers watched an ASMR video, and completed assessments of neuroticism, trait anxiety, and pre- / post-video state anxiety. MANCOVA with Group as the independent measures factor showed that ASMR-experiencers had significantly greater scores for neuroticism, trait anxiety, and video engagement than non-experiencers. Pre-video state anxiety was also significantly greater in the ASMR-experiencers and was significantly attenuated on exposure to the ASMR video, whereas non-experiencers reported no difference in state anxiety pre- and post-video. Thus, watching ASMR alleviated state anxiety but only in those who experienced ASMR. Subsequent mediation analyses identified the importance of pre-existing group differences in neuroticism, trait and (pre-video) state anxiety in accounting for the group difference in the reduction of state anxiety. The mediation analysis further lends support for watching ASMR videos as an intervention for the reduction of acute state anxiety. Future areas for research are discussed.

## Introduction

The Autonomous Sensory Meridian Response (ASMR) is a deeply relaxing sensory phenomenon described as a pleasant tingling sensation originating from the scalp and neck which can

**Funding:** The authors received no specific funding for this work.

**Competing interests:** The authors have declared that no competing interests exist.

spread to the rest of the body [1–3]. ASMR occurs involuntarily and is induced by focusing on auditory, tactile, or visual triggers. Common examples of these triggers include receiving or watching someone else receive personal attention, such as massages or hair brushing, and also from listening to soft sounds such as whispers or tapping [2, 4]. The mechanisms of ASMR are currently unknown, however it has been likened to frisson (musical chills) since both are characterised by a tingling sensation which is invoked by stimuli deemed pleasant by the perceiver [5, 6]. Research incorporating fMRI methodology has shown that multiple brain areas which are activated during frisson, including the nucleus accumbens (NAcc) located in the basal forebrain and which is functionally associated with the reward system, are also activated during ASMR [7]. However, those who experience ASMR sensations insist the phenomena are distinct (see [8]): frisson elicits awe and involves rapidly culminating tingles across the whole body, whilst ASMR elicits relaxation and involves longer-lasting dynamic tingles which originate in the head and neck before spreading to other areas [9]. The perceived difference between the two experiences may be explained by activity in the medial prefrontal cortex (Mpfc) which is heightened during ASMR but reduced during frisson [7, 10].

Whilst currently only a relatively small proportion of the population are thought to be capable of experiencing ASMR, it is certainly gaining popularity, with increasing numbers of aficionados in the online ASMR community [11]. The establishment of this community has resulted in the creation of countless ASMR videos on forums such as YouTube, attracting thousands, or in some instances, millions of subscribers and hits (e.g. [12]; see [2], who list several of the most popular channels). Some of these videos involve simple triggers such as hand movements, tapping on objects or whispering into microphones, whilst others may be more elaborate, simulating situations such as doctor's appointments, makeovers, or haircuts [2, 13]. It is important to emphasise that, due to the subjective nature of ASMR, there is substantial individual variability in which triggers induce the sensation [14]. That is, what causes one individual to experience ASMR might not induce the response in another (see [15] for differential fMRI activity in response to different ASMR triggers).

According to both creators (referred to as ASMRtists) and viewers, ASMR videos promote well-being, relaxation, and sleep. This is not only evident from forums and video comment sections, but is also supported by empirical research findings [11, 16, 17]. In the seminal paper by Barratt and Davis [2], out of a sample of 475 participants, 70% reported watching ASMR videos in order to deal with stress, 82% to aid sleep, and 98% to relax. Since stress, insomnia, and an inability to relax are all typical indications of elevated levels of anxiety [18], this raises the question as to whether ASMR could offer the possibility of a non-conventional intervention in the treatment of anxiety. Clinical anxiety is characterised by continuous or repeating states of fear and/or panic, and with an increasing prevalence of diagnosis [19]. According to the most recent NHS survey [20] there has been a 20% increase in clinical anxiety since 1993, now affecting 18.9% of the population in England. Whilst there are a number of different treatments available for anxiety, they are not always effective for everyone [18] and alternative nonclinical forms of treatment are increasing in popularity [21]. A recent meta-analysis provides substantial empirical evidence for positive effects of mindfulness meditation in the reduction of symptoms of anxiety [22], and emerging literature suggests that ASMR may be mindfully induced [2, 9, 23]. Furthermore, individuals who experience ASMR report higher scores in indices of mindfulness such as the Mindful Attention and Awareness Scale [9, 23, 24]. To date, no known published research has investigated any direct link between experiencing ASMR and reduced anxiety, however psychological and physiological symptoms of anxiety have been shown to decrease after experiencing ASMR [2, 4]. These studies found that experiencing ASMR resulted in increased positive affect and decreased negative affect, along with physiological changes of increased skin conductance and reduced heart rate, similar to those observed

during mindfulness meditation [25]. Therefore, when taking into account the suggested link between ASMR and mindfulness, it is plausible that ASMR may have the same therapeutic applications within an anxiety context.

Due to the recency of ASMR as a subject of empirical investigation, research attention has predominantly focused on the formal descriptive parameters of the phenomenon and the effects it has on those who watch ASMR videos. Consequently, there is limited research investigating traits that characterise individuals who actually experience ASMR. To date there is only a small body of published literature investigating any link between personality traits and the ability to experience ASMR (e.g. [26–28]). Incorporating the Big Five Index (BFI) Fredborg et al. [26] found that ASMR-experiencers scored significantly higher than non-experiencers on neuroticism and openness-to-experience (also see [16, 29]). Greater scores on neuroticism are of particular relevance to the present study as this is often indicative of dispositional determinants of negative mental states [30]. One of the emotion-related sub-domains of the neuroticism scale is anxiety and can be differentiated two ways: *State Anxiety* refers to the refers to the level of moment-to-moment anxiety an individual experiences, whereas *Trait Anxiety* reflects a stable and enduring tendency to experience anxiety and which predicts incidence of clinical anxiety [31, 32]. Research has identified neuroticism to be a predictor of various anxiety disorders including general anxiety disorder (GAD) and major depressive disorder (MDD) ([33]; also see [34] for further discussion). Similar to individuals scoring higher in neuroticism, those with a diagnosis of MDD are more likely to have a clinical anxiety diagnosis [35]. Notably, a significant number of those who are capable of experiencing ASMR suffer from MDD [2]. It is therefore plausible to suggest that if greater levels of neuroticism are found amongst ASMR experiencers, this may predict negative affect and thus an increased disposition for elevated trait and state anxiety in these individuals.

Whilst the impact of ASMR on short-term reduction of depression has been investigated [2], there is no known research to date that has directly investigated the effect of watching ASMR videos and subsequent reduction in state anxiety, and only one known published study that has associated affect-related personality traits of neuroticism as a predictor in experiencing ASMR [26]. Thus, the aim of the present study is threefold: a) to determine whether ASMR experiencers and non-experiences differ on the characteristics of neuroticism, trait, and state anxiety; b) whether exposure to ASMR videos reduces state anxiety in general, or whether this is specific to those who experience ASMR, and c) identify if these putative trait differences mediate any reduction in state anxiety. Due to the limited research and the exploratory nature of the study, the hypotheses are generated cautiously. Firstly, it was hypothesised that the group of individuals who experience ASMR would report higher scores in neuroticism, trait anxiety, and pre-ASMR video state anxiety compared to those who do not experience ASMR, thus providing a metric by which this group can be characterised. Secondly, due to the similarities with mindfulness, those who experienced ASMR would report reduced state anxiety post-exposure to ASMR videos. Finally, following the assumption that ASMR experiencers and non-experiencers may differ in neuroticism, state and trait anxiety, a further exploratory question was pursued, attempting to identify whether these personality factors could mediate any observed change in state anxiety between the groups in response to the ASMR video.

## Method

### Participants

The sample consisted of 64 participants aged from 18 to 58 years ($M$ = 29.55, $SD$ = 11.54); 71.88% identified as *female*, 26.56% as *male*, and 1.56% as *other*. Participants were recruited from online ASMR community forums and YouTube channels, or from social media

**Table 1. Demographics of the total sample, ASMR-experiencers and non-experiencers.**

| | | Total Sample | ASMR Experiencers | Non-experiencers |
|---|---|---|---|---|
| | n | 64 | 36 | 28 |
| Gender | M / F / other | 17 / 46 / 1 | 9 / 26 / 1 | 8 / 20 / 0 |
| Age | range | 18–58 | 18–58 | 18–56 |
| | mean | 29.55 | 30.42 | 28.43 |
| | *SD* | *11.54* | *11.13* | *12.15* |
| Previously seen ASMR Video [a] | n | 38 | 28 | 10 |

Note.

[a] reflects the number of participants answering "yes" to this question.

platforms such as Facebook and Instagram. Participants were categorised as either ASMR-experiencer or ASMR non-experiencer based upon whether they had experienced ASMR previously, and the location of any tingles if applicable; and / or on their responses on the post-video AMSR experience self-report questionnaire which identified whether they had experienced any tingles during the video, and if so, the location of these. ASMR is described as a tingling sensation *originating from the head and/or neck* [1–3, 36] and which we refer to as 'true ASMR'. Therefore the categorisation of participants into the ASMR-experiencer group depended on the self-reported *origin* of the sensation including their head and / or neck. Nineteen participants consistently experienced true ASMR pre- and post-video, whilst 9 participants who self-identified as ASMR-experiencers pre-video did not report tingles post-video. However, as the location of their tingles pre-video corresponded with the definition of true ASMR, they were included in the ASMR-experiencers group ([9, 36]; see the materials and procedure sections). Additionally, 8 participants who had never watched ASMR videos prior to the study, reported true ASMR tingles post watching the video, and were also included in the ASMR-experiencers group (*n = 36*, mean age 30.42, SD 11.13; see [37]). The remaining 28 participants either never experienced any ASMR sensations pre- or post-video, or reported sensations not consistent with true ASMR (i.e. false-positive reporting of AMSR), and were assigned to the non-experiencers group (mean age 28.43, SD 12.15). There was no significant difference in age ($t = 0.61$, $p = 0.498$) or gender ($X^2 = .064$ $p = 0.800$) between the two groups. See Table 1 for full demographics.

## Materials

The online survey was created using Qualtrics (www.qualtrics.com) and consisted of the following: demographic questions, the neuroticism scale from the Big Five Inventory (BFI; [38]), the State-Trait Anxiety Inventory (STAI; [31]), a 5-minute ASMR YouTube video [39], a video engagement questionnaire, and pre- and post-video ASMR-experience questionnaire.

## Neuroticism scale

Scores on neuroticism were assessed using the relevant 8-items taken from the BFI [38]. Example items include statements such as 'I am depressed', 'I am relaxed' and 'I can be moody'. Participants were asked to respond to each item using a 5-point Likert scale from 1 = 'strongly disagree' to 5 = 'strongly agree', with items 2, 5, & 7 reversed scored. The overall neuroticism score was calculated from the mean score of the 8 items, with higher scores indicating greater trait neuroticism.

## State-Trait Anxiety scales

The STAI consists of 40 items: 20 for the State-Anxiety scale (S-Anxiety) and 20 for the Trait Anxiety scale (T-Anxiety). The T-Anxiety scale was used in this study to measure characteristic-level anxiety. Example items include 'I feel like a failure' and 'I feel rested'. The S-Anxiety scale was used to measure the level of momentary anxiety at the current time. Since it detects transitory anxiety at a mental level, it was considered appropriate to detect the effect of ASMR on short-term change in anxiety. Example items include 'I am tense' and 'I feel at ease'. Both tests were answered via a 4-point scale with 1 = 'Very much' and 4 = 'Not at all', where higher scores indicated greater state and trait anxiety. Half of the items in each test were reverse scored, and the overall score for state and trait anxiety was calculated by summing the items for each scale.

## ASMR video

A 5-minute ASMR video by Cynthia Henry ASMR ([39]; *ASMR; 5 Minutes Quick Triggers Ep. 1*) was used for the study https://www.youtube.com/watch?v=RB4wY1-NecI&t=8s. This consisted of a wide range of triggers, such as tapping, scratching, applying make-up, and microphone brushing, presented in rapid succession.

## ASMR video engagement questionnaire

Video engagement was assessed with a bespoke 3-item questionnaire (1: Did this video relax/calm you? 2: Did this video make you uncomfortable? 3: Did this video make you feel sleepy?) based on predominant reasons individuals watch ASMR (see [2]). Participants respond to each question using a 5-point Likert scale where 1 = 'Not at all' and 5 = 'Very much' (Q2 was reverse scored) and with higher scores indicating a more positive experience.

## ASMR experience questionnaire

Presence of tingling sensation alone was not deemed sufficient evidence for the ability to experience ASMR. Since frisson (chills) is common when focus is applied to something pleasant [9], there was a possibility that participants reported experiencing tingles which were not part of ASMR. Due to social distancing restrictions due to the Covid-19 pandemic at the time of data collection, physiological parameters such as skin conductance and heart rate could not be measured in order to ensure ASMR was not mistaken for frisson. Instead, whether the participants truly experienced ASMR or not was determined using a questionnaire. These involved two 'qualifying' questions regarding the location of any tingles. Firstly, participants were asked *'Did you experience a tingling sensation'* which required a Yes / No forced choice response. If they answered *'Yes'* then they were asked *'Where did these tingles start'* to which they responded by typing their response in a text box. Participants completed the questionnaire pre- and post-watching the ASMR video. The inclusion of pre- and post- video assessments was necessary in order to account for individuals who are able to experience ASMR but did not during the video provided, as well as those who had never come across or experienced the phenomenon before the study, but experienced ASMR during the video for the first time.

## Procedure

Participants were directed to the Qualtrics link where the purpose and the procedure of the study was explained and participants could give informed online consent. Participants were asked to find somewhere quiet and wear headphones for the duration of the study to maximise the chance that ASMR was experienced (see [6]). They were presented with the demographic

questions and pre-video ASMR experience questionnaire. Next, they completed the trait anxiety and neuroticism measures, and the pre-video state anxiety questionnaire. They were then asked to follow the on-screen link and watch the 5-minute ASMR video. Following the video, participants completed the post-test measure of state anxiety, followed by the post-video engagement and post-video ASMR experience questionnaires. Participants were debriefed following completion of the survey and directed to online ASMR communities if interested in learning more about the phenomenon. The study took approximately 10–15 minutes to complete, which received ethical approval from the Department of Psychology, Northumbria University Ethics Committee.

### Data treatment

Descriptive statistics were initially carried out to identify Group scores in trait anxiety, neuroticism, state anxiety pre-ASMR video exposure (pre-video), state anxiety post-ASMR video exposure (post-video), and scores on the ASMR video engagement. Following this, correlation and partial correlation analyses controlling for age and gender were carried out on the key variables.

The MANCOVA looked at the differences between the ASMR-experiencer and non-experiencer groups, in terms of neuroticism, trait anxiety, and pre-video exposure state anxiety, controlling for age and gender. The mixed ANCOVA analysis (age and gender as covariates) with Group as the independent measures factor, and pre- versus post-video state anxiety scores as the repeated measures factor examined whether there was a general reduction in state anxiety or whether this was more specific to the ASMR-experiencer group. Following point biserial analyses of the primary relationship between the Group variable and the outcome measure, change in state anxiety, a series of mediation analyses were then carried out. Group was the main predictor, change in state anxiety (pre- vs. post-video values) as the outcome variable, and age and gender as covariates. Neuroticism, trait anxiety, pre-video state anxiety measures were initially employed as the single mediators. In addition, the participants' engagement with the ASMR video acted as a mediator in the final analysis in order to identify the importance of the video engagement in accounting for any group differences in state anxiety change. All analyses were conducted using SPSS v26. The dataset for the current study can be accessed via https://doi.org/10.25398/rd.northumbria.16654846

### Results

Participants were categorized into ASMR-experiencers or non-experiencers according to their answers in response to the *ASMR Experience* questionnaire. Means and standard deviations were calculated for the key variables and are shown in Table 2 below. Table 2 indicates that the ASMR-experiencer group demonstrate higher mean scores in all of these key measures.

**Table 2. Summary of descriptive statistics of ASMR-experiencers and non-experiencers.**

| Group | | Trait Anxiety | Neuroticism | Pre-video State Anxiety | Post-video State Anxiety | Change in State Anxiety | Video Engagement |
|---|---|---|---|---|---|---|---|
| ASMR Experiencer *n = 36* | Mean | 49.17 | 3.41 | 41.75 | 36.58 | 5.17 | 3.46 |
| | SD | 13.29 | 0.63 | 12.82 | 11.87 | 10.83 | 1.12 |
| ASMR Non- Experiencer *n = 28* | Mean | 41.00 | 2.86 | 35.04 | 35.14 | -0.11 | 2.71 |
| | SD | 12.81 | 0.85 | 9.05 | 10.39 | 7.39 | 0.99 |
| Overall | Mean | 45.59 | 3.17 | 38.81 | 35.95 | 2.86 | 3.14 |
| | SD | 13.61 | 0.78 | 11.73 | 11.18 | 9.77 | 1.12 |

**Table 3. Correlations between the key predictor variables, trait anxiety, neuroticism, pre-video state anxiety and video engagement.**

|  | Trait anxiety | Neuroticism | Pre- video state anxiety | ASMR video engagement |
|---|---|---|---|---|
| Trait Anxiety | —— | 0.85*** | 0.75*** | 0.19 |
| Neuroticism | 0.848*** | —— | 0.65*** | 0.20 |
| Pre-video State Anxiety | 0.777*** | 0.70*** | —— | 0.19 |
| ASMR video engagement | 0.26* | 0.27* | 0.22 | —— |

* p<0.05

***p<0.001.

*Note*: The values above the diagonal are zero-order correlations and below the diagonal, partial correlations controlling for age and gender.

In order to explore the relationships between the key variables, correlations and partial correlation analyses were carried out. The findings are shown in Table 3 below and indicates significant positive relationships between trait anxiety, neuroticism, and pre-video state anxiety ($p < 0.001$). In addition, controlling for age and gender continued to reveal significant positive correlations between trait anxiety, neuroticism, and ASMR video engagement ($p < 0.001$).

## Comparing traits between the groups

In order to test the first hypothesis, trait anxiety, neuroticism, pre-video state anxiety and ASMR video engagement scores were compared between ASMR-experiencers and non-experiencers. A one-way multivariate analysis of variance controlling for age and gender (MANCOVA) determined that there was a statistically significant difference in overall character trait scores between the groups. Using Wilk's lambda, the MANCOVA revealed a significant effect associated with Group, $\lambda = 0.79$, $F(4,56) = 3.72$, $p = 0.009$, partial $\eta^2 = .210$, Box's $M = 19.64$, $p = 0.052$. Subsequent ANOVA analyses showed a significant Group effect with trait anxiety, $F(1,59) = 5.51$, $p = 0.022$, partial $\eta^2 = .085$; neuroticism, $F(1,59) = 9.62$, $p = 0.003$, partial $\eta^2 = .140$; and pre-video state anxiety, $F(1,59) = 4.88$, $p = 0.032$, partial $\eta^2 = .076$; ASMR video engagement, $F(1, 59) = 7.80$, $p = 0.006$, partial $\eta^2 = .119$. With all of the measures there was significantly greater trait anxiety, neuroticism, pre-video state anxiety and ASMR video engagement scores observed in the ASMR-experiencer group compared to the non-experiencer group.

In order to explore the second hypothesis, a mixed 2-way ANCOVA was carried out, with Group as the independent measures factor and pre- and post-video state anxiety scores as the repeated measures factor. Age and gender were employed as co-variates. The ANCOVA revealed that there was a main effect of the repeated measures factor, pre-video vs post-video, $F(1,62) = 4.48$, $p = 0.038$, partial $\eta^2 = .067$ with significantly greater state anxiety prior to exposure to the video. There was no significant Group difference, $F(1,62) = 2.49$, $p = 0.119$, partial $\eta^2 = .039$. However, the interaction was significant, $F(1,62) = 4.87$, $p = 0.031$, partial $\eta^2 = .073$.

In order to clarify the significant interaction between the independent variables, a series of paired and independent sample t-tests were conducted. Firstly, paired samples t-tests were conducted to compare the pre-video state anxiety scores to the post-video state anxiety scores for each group. The tests found that the ASMR-experiencer group had a significantly lower state anxiety scores post-video ($M = 36.58$, $SD = 11.87$) than pre-video ($M = 41.75$, $SD = 12.82$), $t(35) = 2.86$, $p = 0.007$, $d = 0.48$. In contrast, there was no difference between the pre- ($M = 35.04$, $SD = 9.05$) and post-video scores for the non-experiencer group ($M = 35.14$, $SD = 10.39$), $t(27) = -0.08$, $p = 0.939$, $d = -0.02$. Secondly, independent sample t-tests were run to determine whether there was a statistically significant effect of Group on the pre-video and post-video state anxiety scores. The test found significantly higher pre-video state anxiety

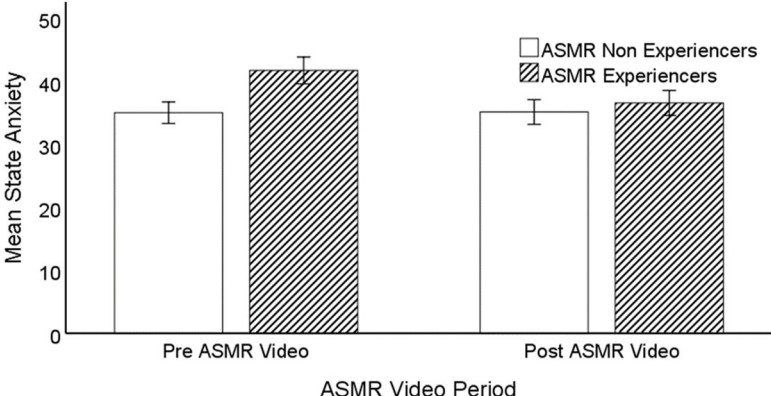

**Fig 1. Mean state anxiety scores of ASMR-experiencer and non-experiencer groups, pre- versus post-video exposure.** Note: bars are standard errors.

scores in the ASMR-experiencer group (*M* = 41.75, *SD* = 12.82) than the non-experiencer group (*M* = 35.04, *SD* = 9.04), *t(61.49) = 2.45, p = 0.017, d = 0.58*. However, there was no significant difference between the ASMR-experiencer group (*M* = 36.58, *SD* = 11.87) and the non-experiencer group (*M* = 35.14, *SD* = 10.39) when comparing post-video state anxiety scores, *t(62) = 0.51, p = 0.613, d = 0.13* (see Fig 1).

In order to investigate the final hypothesis, a series of mediation analyses [40] were carried out. The initial correlation analysis between Group and change in state anxiety revealed $r_{pb}$ = *-.270, p = .037*. The outcomes are shown in Fig 2.

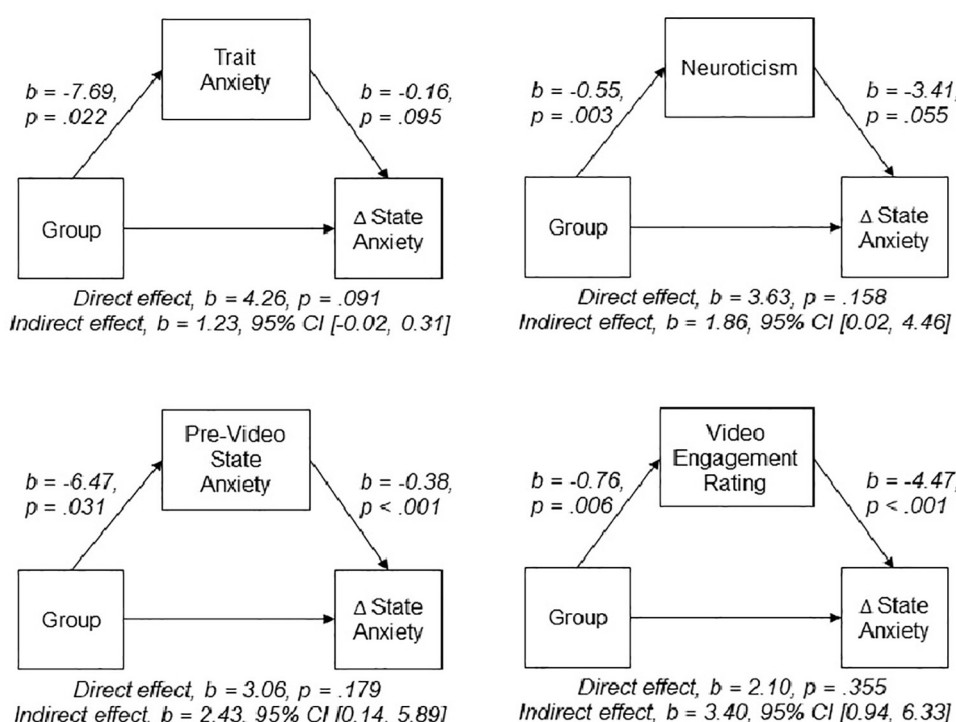

**Fig 2. The contribution of trait anxiety, neuroticism, pre-video state anxiety and ASMR video engagement as mediators of the Group-change in state anxiety relationship.**

Fig 2 shows that the insertion of each of the mediators into the Process model makes the direct relationship between Group and change in state anxiety non-significant, with neuroticism, pre-video state anxiety, and ASMR video engagement rating each acting as significant mediators.

### Analysis summary

The findings suggest that significant differences in personality characteristics were present between the two groups in trait anxiety, neuroticism, and in pre-video state anxiety. In addition, there was a significant group difference in ASMR video engagement. The correlations and partial correlations revealed significant relationships between trait anxiety, neuroticism, and pre-video state anxiety and some significant relations of these variables with ASMR video engagement. The mixed ANOVA with pre- versus post-video exposure (state anxiety scores) and Group revealed a significant interaction, with only the ASMR-experiencer group evidencing a significant reduction in state anxiety. Finally, mediation analyses revealed that neuroticism, pre-video state anxiety, and the ASMR video engagement scores all acted as significant mediators, with the direct relationship between Group and change in state anxiety no longer being significant.

## Discussion

The primary aim of the study was to investigate whether the ability to experience ASMR is associated with higher levels of neuroticism, trait anxiety, and state anxiety. The second aim was to identify whether watching ASMR videos helps reduce state anxiety in general, or whether any benefit was subject to actually experiencing the phenomenon. We also investigated whether any difference in personality characteristics between the groups could be mediating the difference in the impact of the video on state anxiety.

The results upheld our first hypothesis, that individuals who are able to experience ASMR have significantly greater neuroticism, state anxiety, and trait anxiety scores compared to non-experiencers. The significantly greater neuroticism scores observed in the ASMR-experiencers is consistent with the limited prior studies which have investigated personality traits in ASMR-experiencers (e.g. [16]—though their group differences were rendered non-significant with multiple comparisons, [26, 28]). Neuroticism and trait anxiety are known to be strongly linked since neuroticism describes a predisposition to negative emotional states such as anxiety [30]. Thus, the results here suggests that those with the ability to experience ASMR are more likely to experience negative emotional states and have a propensity for trait anxiety. This relationship was confirmed by the strong positive correlation between the scores of neuroticism and trait anxiety. The results also supported the second hypothesis, as only the ASMR-experiencers reported a decrease in state anxiety as a result of watching the ASMR video. This result was driven by significantly greater pre-video state anxiety in the ASMR-experiencer group compared to the non-experiencers. In contrast, there was no difference in pre- and post-video state anxiety in the non-experiencers group. This suggests that ASMR-experiencers also have greater predisposition for baseline state anxiety, which can be alleviated by watching ASMR videos. This concurs with the prior observation of a relationship between trait and state anxiety where high trait anxiety is indicative of greater baseline level of state anxiety [31].

When considering the ASMR video engagement ratings, correlational analyses found both trait anxiety *and* neuroticism scores were significantly positively correlated with video engagement scores, when age and gender were controlled for, suggesting that the severity of disposition for negative affect and anxiety are related to enjoyment of ASMR videos. Analyses of group differences found ASMR-experiencers gave significantly higher video engagement

scores than the non-experiencers. This supports several studies which suggest ASMR is only enjoyable for individuals who experience the phenomenon and may even be unenjoyable for non-experiencers [2, 4, 9, 11]. Furthermore, video engagement scores were strongly correlated with change in state anxiety, suggesting that the more positive the ASMR engagement, the greater the reduction in short term anxiety. Similar results have been reported previously [2], where ASMR experiencers who had been diagnosed with depression experienced the greatest decrease in depressive symptoms as a result of watching an ASMR video, though caution needs to taken with this comparison as depression and anxiety are discrete types of psychopathology. However, when considering the comparisons made between mindfulness and ASMR, in order to experience ASMR, mindful focusing is required [9, 23, 24, 41]. As mindfulness has been shown to reduce anxiety (e.g. [22, 42]; but also see [43] for alternative findings), it is unsurprising that ASMR has similar positive effects. Combined, this lends further support for watching ASMR videos as an intervention for the reduction of acute state anxiety, but only in those with a propensity to experience ASMR.

However, the results from the mediation analysis suggest alternative interpretations need to be considered. This analysis found that neuroticism, pre-video state anxiety, and video engagement each acted as significant mediators resulting in the relationship between Group and change in state anxiety being rendered non-significant. Therefore, this leads us to question whether ASMR *could* be considered as an intervention even for those who do not actively watch ASMR videos and / or have experienced ASMR tingles but do have elevated levels of neuroticism and anxiety. This was implied by the findings of a recent study employing electro-encephalography (EEG) methodology with a non-clinical sample [44]. Participants completed a difficult mental task while either listening to an ASMR audio of choice or in silence. The authors found that, when completing the mentally demanding task was accompanied with an ASMR audio, the alpha and beta frequency band levels increased to levels commensurate of resting states (also see [45]). Greater alpha and beta power values are also observed during mindfulness mediation [46–48], thus the study by Ohta and Inagaki [44] provides a seminal indicator of the functional significance of ASMR, as a means to aid relaxation and mental stress, even during demanding cognitive processing. There are two notable implications from this study. Firstly, it re-emphasises the importance of using the participants' choice of trigger, which was lacking in the current study (see [15]). Secondly, the comparable high levels of alpha whilst completing the mentally demanding task suggest that ASMR may have a clinical application even under conditions of high cognitive load. When considering that individuals typically watch ASMR under quiet, relaxing conditions with focused attention (similar to those when practicing mindfulness mediation), the findings of Ohta and Inagaki [44] suggest that the benefits of ASMR may be evident under alternative conditions. This work clearly requires replication and should be supplemented with biomarkers such as galvanic skin response (GSR) and heart rate, as changes in these are indicative of reduced anxiety and have been observed in response to ASMR exposure [4, 11, 49].

The present study is not without limitations. We cannot discount that there was strong selection bias amongst the sample group. Most of the ASMR-experiencer group consisted of participants who were recruited online from ASMR forums or social media comment sections. Thus, the recruitment method resulted in 28 out of 36 ASMR-experiencer participants having prior knowledge and experience of ASMR. That is, since the ASMR-experiencer group consisted mainly of ASMR viewers, we cannot be certain of whether the effects seen are representative of all ASMR-capable individuals or whether they just belong to individuals who are actively seeking out ASMR videos. However, 8 participants allocated to the ASMR-experiencer group, based on the location of their tingles post-video, had never been exposed to ASMR previously [37]. This emphasises the importance of recruiting 'ASMR-naïve' participants in future

studies. The benefit of recruiting an 'ASMR-naïve' sample would enable researchers to clarify whether the proposed character traits were indeed reflective of *all* ASMR-experiencers, or just the experiencers who view ASMR regularly. Certainly, inclusion of ASMR-naïve individuals prone to high anxiety warrants further investigation.

Nine participants who identified as experiencing ASMR prior to the study and were allocated to the ASMR-experiencers group, did not experience ASMR tingles during the study. This likely reflects the individual differences in which ASMR triggers that precipitate a response [2, 15, 26]. To try and counter this, a video with as many triggers as possible was selected for the study; however, the brief duration of the video meant each trigger was also relatively short. Therefore, it is plausible that some ASMR-capable individuals simply did not experience ASMR tingles during the video used in the current study. This might be overcome by including a video with a large range of longer-lasting triggers. Alternatively, future studies would benefit from including an option for participants to choose which type of ASMR videos they want to watch (e.g. [15, 44]), especially as stimulus familiarity may also be implicit in greater experience of ASMR as is observed with musically induced frisson [50]. We also need to consider that nine participants allocated to the non-experiencer group *did* report tingles either pre-and / or post-video, however the location of these was not consistent with 'true' ASMR which originates in the head and / or neck. This emphasises the importance of more fully identifying participants' usual ASMR experiences as well as those during experimental investigation in order to rule out false-positive reporting of the experience (see [36] for a discussion).

The measures used in the current study are also not without question. Whilst use of the STAI is widespread, its ability to discriminate anxiety from depression has been questioned (e.g. [51]). Thus, future research might consider an alternative measure such as the State-Trait Inventory for Cognitive and Somatic Anxiety (STICSA; [52]; for a discussion see [53]). Another issue was the brevity of the video engagement questionnaire, therefore future studies should aim to use more psychometrically validated measures such as the ASMR-15 inventory [54] or the ASMR-Experience Questionnaire (AEQ; [36]), though of note the video engagement questionnaire used here *did* contribute significantly to the mediation model in Fig 2.

In summary, the primary results support the hypothesis that ASMR-capable individuals score higher in neuroticism and trait anxiety than non-experiencers. This suggests that ASMR-experiencers are particularly prone to experiencing negative emotional states as well as anxiety disorders. The main findings also provide evidence in support of the second hypothesis which proposed exposure to ASMR videos would reduce state anxiety in ASMR-experiencers but not non-experiencers, indicative that ASMR has positive effects in those who are capable of experiencing the phenomenon. Furthermore, investigation of the interaction between timing and group revealed that ASMR-experiencers have higher baseline levels of state anxiety than average, where scores are reduced to a typical level as a result of watching ASMR. Though there were certain limitations in the methodology of the study, these primary outcomes provide strong support of the prospect that ASMR has the potential to be both effective and suitable as a clinically relevant anxiety treatment. Non-clinical future studies should direct investigations towards EEG and fMRI studies, particularly in order to verify the potential mechanism behind ASMR's anxiety reducing capabilities. Another important avenue would be to investigate the possibility that ASMR *viewers* are characteristically different to ASMR-capable individuals with no prior experience of online ASMR. Whilst it seems logical that those who experience ASMR report greater video engagement ratings, the results here suggest that ASMR has the potential to have anxiety-reducing effects in general; in particular as our data imply that individuals with high trait anxiety levels are more likely to be able to experience ASMR than those lower in trait anxiety. We need to consider that the targeted group in the

current study may have a particular predisposition to seek out and experience ASMR. However, the pattern of reduced state anxiety, along with the results of the mediation analyses, suggest that ASMR could be employed as a clinical intervention in general, targeting any group of individuals who have elevated levels of neuroticism or anxiety.

## Author Contributions

**Conceptualization:** Charlotte M. Eid, Joanna M. H. Greer.

**Data curation:** Charlotte M. Eid, Colin Hamilton.

**Formal analysis:** Charlotte M. Eid, Colin Hamilton, Joanna M. H. Greer.

**Investigation:** Charlotte M. Eid.

**Methodology:** Charlotte M. Eid, Joanna M. H. Greer.

**Project administration:** Charlotte M. Eid.

**Supervision:** Joanna M. H. Greer.

**Validation:** Colin Hamilton, Joanna M. H. Greer.

**Visualization:** Charlotte M. Eid.

**Writing – original draft:** Charlotte M. Eid.

**Writing – review & editing:** Colin Hamilton, Joanna M. H. Greer.

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
