## [Decision Letter · Decision Letter 0]

18 Aug 2021

PONE-D-21-21005

Untangling the tingle: Investigating the association between the Autonomous Sensory Meridian Response (ASMR), neuroticism, and trait & state anxiety

PLOS ONE

Dear Dr. Greer,

Thank you for submitting your manuscript to PLOS ONE. After careful consideration, we feel that it has merit but does not fully meet PLOS ONE’s publication criteria as it currently stands. Therefore, we invite you to submit a revised version of the manuscript that addresses the points raised during the review process.

We really appreciate for your interesting and important manuscript for PLOS ONE. Both reviewers admitted the importance of your study, but they also pointed some important problems in your manuscript, especially about the analysis of the results. 

We are looking forward to reading your revised manuscript soon.

We look forward to receiving your revised manuscript.

Kind regards,

Nobuyuki Sakai, Ph.D.

Academic Editor

PLOS ONE

Reviewers' comments:

Reviewer's Responses to Questions

**Comments to the Author**

1. Is the manuscript technically sound, and do the data support the conclusions?

Reviewer #1: Partly

Reviewer #2: Partly

2. Has the statistical analysis been performed appropriately and rigorously? 

Reviewer #1: No

Reviewer #2: I Don't Know

3. Have the authors made all data underlying the findings in their manuscript fully available?

Reviewer #1: Yes

Reviewer #2: Yes

4. Is the manuscript presented in an intelligible fashion and written in standard English?

Reviewer #1: Yes

Reviewer #2: Yes

5. Review Comments to the Author

Reviewer #1: The authors investigated whether participant anxiety was reduced by watching an ASMR video. In this study, 36 ASMR experiencers and 28 ASMR non-experiencers watched a 5-min ASMR video and their neuroticism, trait anxiety, and pre-/post-video state anxiety were assessed. Scores of neuroticism, trait anxiety, and video experience were greater for ASMR experiencers than for non-experiencers. The reduction of pre- and post-video state anxiety was determined for ASMR experiencers, but was not for non-experiencers. Mediation analyses showed that group differences in neuroticism and trait anxiety accounted for the group difference in the reduction of state anxiety. The authors suggest that ASMR experiencers have a predisposition to experience negative emotional states, as well as anxiety disorders.

The literature is reviewed in a clear and concise manner. As noted by the authors, an early study showed that watching ASMR stimuli can improve participant mood for several hours, suggesting that depressed people derive the greatest benefit from engaging in ASMR (Barratt & Davis, 2015). A recent study demonstrated that sensitivity to ASMR stimuli is correlated with participant anxiety state (Koumura et al., 2021, Q J Exp Psychol), although the authors did not cite this paper. What is the critical difference between anxiety and mood? How are the present and previous findings related? The authors should discuss these issues.

The authors claimed selection bias of participants as a limitation of this study. They excluded 10 participants as ASMR experiencers and included 8 ASMR non-experiencers among ASMR experiencers. Is it possible to derive generalized principles when using such methods? In addition, the authors used only one ASMR stimulus. Indeed, the stimulus consisted of some situations (e.g., brushing, tapping, and scratching), but was rapidly changing. When the authors checked the participant ASMR experiences, I consider the following questions inappropriate: Did this video relax/calm you? or did this video make you feel sleepy? (p. 9) It is better to ask participants directly whether they really experienced a tingling sensation.

The authors performed mediation analyses (Fig. 2), but did not show the correlation between predictor and outcome: Group and change in an anxiety state. If the authors do not find a significant correlation between predictor and outcome, then it would be pointless to carry out a mediation analysis. Thus, they should explicitly show the correlation and then discuss effects of mediators on outcomes.

Reviewer #2: The manuscript presents an interesting study in which participants who report to experience ASMR are compared to participants who report to not experience ASMR on general levels of neuroticism and trait anxiety, as well as on how they report state anxiety before and after watching an ASMR video. Previous literature appears sufficiently explored and the study is placed in relevant context. Research questions and hypotheses are presented in a clear fashion. The manuscript is structured appropriately and reads well. However, due to two major issues I recommend major revision.

Major issues:

1. In the method section it is mentioned that participants were categorized as either ASMR-experiencers or non-experiencers based on whether they had previously viewed ASMR videos and on their answers to the video and ASMR experience questionnaires. Based on the text, however, it is unclear how these questionnaires were used to determined whether someone was an ASMR-experiencer. The distinction between ASMR-experiencers and non-experiencers needs to be explained better. I think it would also be beneficial to attach all the questionnaires to the appendix.

2. On line 105-107 it is stated that the first research question was to determine whether ASMR-experiencers and non-experiencers differ on neuroticism, trait, and state anxiety, and the second research question was to determine whether exposure to ASMR videos reduces state anxiety in general or whether it is specific to those who experience ASMR. Based on the text in the participants section on pages 6-7 it appears that 38 participants reported to watch ASMR videos and experience sensations in relation to them, however, 10 of those participants were assigned to the non-experiencer group because they did not experience ASMR tingles during the study. To my understanding, these participants self-identified as ASMR-experiencers. I think assigning participants who report to experience ASMR into a non-ASMR group because they did not experience tingles during the experiment is incorrect and contradicts the analyses conducted, particularly those pertaining to the first two research questions. Participants who report to experience ASMR should not be included in the non-experiencer group.

I would like to thank the authors for an interesting manuscript about an interesting topic. If the analyses are redone with the above group adjustments I would be happy to reconsider.

6. PLOS authors have the option to publish the peer review history of their article (what does this mean?). If published, this will include your full peer review and any attached files.

Reviewer #1: **Yes: **Hirohito M. Kondo

Reviewer #2: No

---

## [Author Response · Author response to Decision Letter 0]

24 Sep 2021

Reviewer #1: 

1: The literature is reviewed in a clear and concise manner. As noted by the authors, an early study showed that watching ASMR stimuli can improve participant mood for several hours, suggesting that depressed people derive the greatest benefit from engaging in ASMR (Barratt & Davis, 2015). A recent study demonstrated that sensitivity to ASMR stimuli is correlated with participant anxiety state (Koumura et al., 2021, Q J Exp Psychol), although the authors did not cite this paper. What is the critical difference between anxiety and mood? How are the present and previous findings related? The authors should discuss these issues.

Response: We thank Reviewer 1 for drawing our attention to this paper which was not available at the time of submission of our manuscript and have added this in the development of our arguments (lines 85, 355). Whilst we agree that there are critical differences between mood and anxiety, due to the novelty of ASMR as a focus of empirical research along with our specific research question, we do not feel that detailed discussion is of such relevance to our paper though we welcome future consideration of this.

2: The authors claimed selection bias of participants as a limitation of this study. They excluded 10 participants as ASMR experiencers and included 8 ASMR non-experiencers among ASMR experiencers. Is it possible to derive generalized principles when using such methods? In addition, the authors used only one ASMR stimulus. Indeed, the stimulus consisted of some situations (e.g., brushing, tapping, and scratching), but was rapidly changing. When the authors checked the participant ASMR experiences, I consider the following questions inappropriate: Did this video relax/calm you? or did this video make you feel sleepy? (p. 9) It is better to ask participants directly whether they really experienced a tingling sensation.

Response: On reviewing our text, we can see that greater clarification is required regarding the criteria in the allocation of participants to the ASMR-experiencer or non-experiencer groups. Group allocation was based on participants’ responses to both pre-study and post-study video ASMR experience questionnaires, and specifically based on the location of the origin of the tingles, which had to be either the head and / or neck. Group allocation was not based on responses to the ASMR video experience questionnaire (which we have now renamed ASMR video engagement questionnaire as we feel this provides greater distinction between the two measures). This text has now been removed (see line 128-129). We agree that exclusion and inclusion of participants to these groups was also not clear and have provided greater clarity below in response to Reviewer 2’s similar comments. The manuscript has been amended to clarify group allocation (see lines 127-143) and also the elaborated data set which is available from https://doi.org/10.25398/rd.northumbria.16654846. 

We also agree that the ASMR video engagement questionnaire was not a sufficient measure in general and we have already acknowledged this in our discussion section (lines 450-457) where we emphasised that more robust measures such as the ASMR-15 Inventory (Roberts et al., 2019) or AEQ (Swart et al., 2021) should be incorporated in future research.

3: The authors performed mediation analyses (Fig. 2), but did not show the correlation between predictor and outcome: Group and change in an anxiety state. If the authors do not find a significant correlation between predictor and outcome, then it would be pointless to carry out a mediation analysis. Thus, they should explicitly show the correlation and then discuss effects of mediators on outcomes.

Response: This correlation has been added on lines 315-316. 

Reviewer #2: 

1. In the method section it is mentioned that participants were categorized as either ASMR-experiencers or non-experiencers based on whether they had previously viewed ASMR videos and on their answers to the video and ASMR experience questionnaires. Based on the text, however, it is unclear how these questionnaires were used to determined whether someone was an ASMR-experiencer. The distinction between ASMR-experiencers and non-experiencers needs to be explained better. I think it would also be beneficial to attach all the questionnaires to the appendix.

Response: We agree with Reviewer 2’s comments here and similar were raised by Reviewer 1. We have addressed this above in response to R1 and below in response to R2’s 2nd comment, where we outline in detail how participants were allocated to either the ASMR-experiencer or non-experiencer groups. To clarify, the ASMR video experience (now renamed video engagement) questionnaire was not used to classify group allocation and this text has been removed. 

2. On line 105-107 it is stated that the first research question was to determine whether ASMR-experiencers and non-experiencers differ on neuroticism, trait, and state anxiety, and the second research question was to determine whether exposure to ASMR videos reduces state anxiety in general or whether it is specific to those who experience ASMR. Based on the text in the participants section on pages 6-7 it appears that 38 participants reported to watch ASMR videos and experience sensations in relation to them, however, 10 of those participants were assigned to the non-experiencer group because they did not experience ASMR tingles during the study. To my understanding, these participants self-identified as ASMR-experiencers. I think assigning participants who report to experience ASMR into a non-ASMR group because they did not experience tingles during the experiment is incorrect and contradicts the analyses conducted, particularly those pertaining to the first two research questions. Participants who report to experience ASMR should not be included in the non-experiencer group.

I would like to thank the authors for an interesting manuscript about an interesting topic. If the analyses are redone with the above group adjustments I would be happy to reconsider.

Response: Thank you for your positive comments regarding our manuscript. We have revisited the full data set and identified that the wording of the allocation of the participants to either the ASMR-experiencers or non-experiencers groups was not clearly documented. However, the error here is in the miswording of the manuscript and not in the mis-categorisation of the participants. Participants were assigned to the ASMR-experiencers group on the basis of experiencing ‘true’ ASMR either pre-video, post-study video, or both. ‘True’ ASMR is experienced when the tingles originate from the head and / or neck (e.g. Tihanyi et al., 2018). To clarify, categorisation of the participants in the study was as follows:

A: Participants who experienced true ASMR pre- and / or post-video: n = 19

B: Participants who experienced true ASMR pre-video but not post-video: n = 9

C: Participants who had never previously experienced ASMR but experienced true ASMR post-video: n = 8

D: Participants who had never previously experienced ASMR prior to participating in the study, and post-video by reporting either no tingles, or tingles that did not originate from the head and / or neck: n = 28

The participants allocated to the ASMR-experiencers group consisted of participants from categories A, B, and C (n=36). The non-experiencers group were participants in category D (n=28). The manuscript has been amended accordingly to clarify the group allocation (lines 127-143). To ensure that the issue related to wording only and not mis-categorisation of the participants, we re-ran all the analyses on the above group re-allocation and observed exactly the same results as we reported in the originally submitted manuscript. The elaborated data set which includes the breakdown of these four categories has been uploaded to the Northumbria University Figshare repository (https://doi.org/10.25398/rd.northumbria.16654846). Relevant comments in the discussion have either been removed or rephrased (see lines 437-448).

---

## [Decision Letter · Decision Letter 1]

8 Nov 2021

PONE-D-21-21005R1Untangling the tingle: Investigating the association between the Autonomous Sensory Meridian Response (ASMR), neuroticism, and trait & state anxietyPLOS ONE

Dear Dr. Greer,

Thank you for submitting your manuscript to PLOS ONE. After careful consideration, we feel that it has merit but does not fully meet PLOS ONE’s publication criteria as it currently stands. Therefore, we invite you to submit a revised version of the manuscript that addresses the points raised during the review process.

Thank you for your resubmitting. Your manuscript is well written and includes important findings in this area. Before accepting your manuscript, I should ask you to address minor points by reviewer 2.  I am so exciting your paper would be published in PlosOne. 

We look forward to receiving your revised manuscript.

Kind regards,

Nobuyuki Sakai, Ph.D.

Academic Editor

PLOS ONE

Journal Requirements:

Reviewers' comments:

Reviewer's Responses to Questions

**Comments to the Author**

1. If the authors have adequately addressed your comments raised in a previous round of review and you feel that this manuscript is now acceptable for publication, you may indicate that here to bypass the “Comments to the Author” section, enter your conflict of interest statement in the “Confidential to Editor” section, and submit your "Accept" recommendation.

Reviewer #1: All comments have been addressed

Reviewer #2: (No Response)

2. Is the manuscript technically sound, and do the data support the conclusions?

Reviewer #1: Yes

Reviewer #2: Partly

3. Has the statistical analysis been performed appropriately and rigorously? 

Reviewer #1: Yes

Reviewer #2: I Don't Know

4. Have the authors made all data underlying the findings in their manuscript fully available?

Reviewer #1: Yes

Reviewer #2: Yes

5. Is the manuscript presented in an intelligible fashion and written in standard English?

Reviewer #1: Yes

Reviewer #2: Yes

6. Review Comments to the Author

Reviewer #1: (No Response)

Reviewer #2: I would like to thank the authors for providing a revised version based on the previous comments. However, I find that the revisions made by the authors thus far were insufficient.

My first issue is related to how the authors differentiate 'true' ASMR-experiencers from false-positives:

On line 188-193 it is mentioned that 'qualifying' questions regarding the location of any tingles was used to assess whether participants truly experienced ASMR. It is unclear to me what these questions where. Based on the text, I understand that only the origin of the sensation was used to make this distinction. This can be done with one question, however, the authors mention that several questions were used. I think it would be important to specify exactly what questions were used to make this distinction.

Furthermore, if the origin of the ASMR sensations was used as the sole criterion to differentiate true ASMR experience from false positives, I think it is not clear from the manuscript why this is sufficient. On lines 197-199 the authors refer to the Fredborg et al. (2017) study and claim that they used a similar 'qualifying' questions regarding the location of tingles. Upon examining the referred study, however, I do not find any reporting that would imply that a similar method was used. To my understanding, the qualifying questions mentioned in Fredborg et al. (2017) refer to their ASMR checklist, which does not seem to include questions regarding sensation location.

To further contradict the use of only the origin of the tingling sensations to differentiate true ASMR from false positives, Barrat & Davis (2015), for example, report that only 63% of their participants reported a consistent origin and 41% of those participants reported the head and 29% the shoulders as the origin. In addition, Swart et al. (2021), who are also referred to in the manuscript, report that false positive ASMR-responders appear to differ most from actual ASMR-responders in that their sensations do not seem to emphasize the head as the most prominent origin of their tingles and typically report them as unpleaseant and not calming.

Therefore, I think the manuscript needs to make more clear how 'true' ASMR experience was determined and clearly specify whether and how the method used is supported by the literature, including further discussion of potential limitations (e.g., might mistaking 'true'. ASMR experiencers as false positives be possible?).

My second (somewhat related) issue is with group assignment and how it is explained:

Although revision to the participants section has been made, I still find the way group assignment is explained not clear. Also, the term 'true' ASMR is mentioned for the first time on line 132 but has not been defined in the preceding text. Since it seems to be the case that experiencing 'true' ASMR is used as the main criterion for group assignment, its meaning should first be explained clearly.

Based on the revised text and the authors responses, it is my understanding that the non-ASMR-experiencer group includes participants who do not report to experience ASMR and also participants who report to experience ASMR but whose experience is not considered 'true' ASMR by the authors. I think that the authors need to explain and motivate their choices for group assignment in the manuscript clearer and in more detail.

Minor issue:

On lines 411-414, it is written that this is the first study to assess whether participants were ASMR-experiencers post-video as well as pre-video. I do not understand what this means. Previous studies have also used questionnaires to 'confirm' or validate participants' ASMR experiencers by asking participants if they consider themselves ASMR experiencers and whether they experienced ASMR while watching a video. How is the method used here different from such studies?

I still think that the manuscript is interesting and publishing it has value. If group assignment and 'true' ASMR is properly defined and explained by the authors, and further limitations are discussed, I would recommend that it be accepted for publication.

7. PLOS authors have the option to publish the peer review history of their article (what does this mean?). If published, this will include your full peer review and any attached files.

Reviewer #1: No

Reviewer #2: No

---

## [Author Response · Author response to Decision Letter 1]

10 Dec 2021

Reviewer #2: I would like to thank the authors for providing a revised version based on the previous comments. However, I find that the revisions made by the authors thus far were insufficient.

1a: My first issue is related to how the authors differentiate 'true' ASMR-experiencers from false-positives:

On line 188-193 it is mentioned that 'qualifying' questions regarding the location of any tingles was used to assess whether participants truly experienced ASMR. It is unclear to me what these questions where. Based on the text, I understand that only the origin of the sensation was used to make this distinction. This can be done with one question, however, the authors mention that several questions were used. I think it would be important to specify exactly what questions were used to make this distinction.

Response:

The participants were asked two questions: a: ‘Did you experience a tingling sensation?’ to which they responded ‘yes’ or ‘no’. If the response was ‘yes’ they were asked b: ‘Where did these tingles start’ to which they responded accordingly in a text box. This text clarifying this has been included in lines 197-199. This enabled us to identify false positives – these data are available in the accompanying dataset. 

1b: Furthermore, if the origin of the ASMR sensations was used as the sole criterion to differentiate true ASMR experience from false positives, I think it is not clear from the manuscript why this is sufficient. On lines 197-199 the authors refer to the Fredborg et al. (2017) study and claim that they used a similar 'qualifying' questions regarding the location of tingles. Upon examining the referred study, however, I do not find any reporting that would imply that a similar method was used. To my understanding, the qualifying questions mentioned in Fredborg et al. (2017) refer to their ASMR checklist, which does not seem to include questions regarding sensation location.

Response:

Thank you for identifying this. We agree that these qualifying questions were not used in the Fredborg et al. (2017) paper and this wording was included in the manuscript in error. We have removed this text.

1c: To further contradict the use of only the origin of the tingling sensations to differentiate true ASMR from false positives, Barrat & Davis (2015), for example, report that only 63% of their participants reported a consistent origin and 41% of those participants reported the head and 29% the shoulders as the origin. In addition, Swart et al. (2021), who are also referred to in the manuscript, report that false positive ASMR-responders appear to differ most from actual ASMR-responders in that their sensations do not seem to emphasize the head as the most prominent origin of their tingles and typically report them as unpleaseant and not calming.

Response:

The paper by Barratt and Davis (2015) states that ASMR is typically characterised by tingles that originate from the head ‘to achieve a tingling, static-like sensation widely reported to spread across the skull and down the back of the neck (Taylor, 2014)’ and this distinction is reported repeatedly within the ASMR literature. This classification enables researchers to dissociate what we have referred to as ‘true’ ASMR from frisson (which is a different sensation e.g. piloerection) and originates in other parts of the body. We have been clear about this in the manuscript – see lines 190-192. Whilst we disagree with Barratt and Davis’ inclusion of the above mentioned 29% as ASMR-experiencers, theirs was a seminal research paper in the area, however knowledge and understanding has advanced since then. There will no doubt always be a debate when it comes to subjective measures regarding what ‘true’ ASMR experience actually is. However, in light of the comparisons with frisson, we feel the need to dissociate this. By stating that participants can only be allocated to the ASMR-experiencer group if they experienced tingles originating in the head and / or neck enables us to make that distinction and is consistent with definition in much of the literature. Whilst we respectfully appreciate that some researchers are happy to include sensations originating from other areas of the body as experiencing AMSR, in our opinion, to be classified as an ASMR-experiencer, the sensations must originate from the head and / or neck and we feel we have been very clear describing this. 

With regard to R2 comments about the Swart et al. (2021) paper, these authors stated that ‘An individual would be deemed a false positive when they report experiencing something, however that something does not align to the hallmark features of ASMR (e.g., pleasant, calming, head-dominant tingles (our emphasis); henceforth termed False- Positive)’. In our manuscript we have been clear that if a sensation does not originate in the head and / or neck, these participants will be allocated to the non-ASMR experiencer group (see lines 130-140 which now includes the word ‘False-Positive’ in order to define this as per R2 comment 2 below). Therefore, we feel that R2’s latter comments here actually concur with our definition. 

1d: Therefore, I think the manuscript needs to make more clear how 'true' ASMR experience was determined and clearly specify whether and how the method used is supported by the literature, including further discussion of potential limitations (e.g., might mistaking 'true'. ASMR experiencers as false positives be possible?).

Response:

The first point has been addressed. Whilst we would welcome further debate in the literature regarding true ASMR / false positive reporting we do not feel further elaboration is relevant to the focus of this paper. We have directed the readers to relevant literature.

2: My second (somewhat related) issue is with group assignment and how it is explained:

2a: Although revision to the participants section has been made, I still find the way group assignment is explained not clear. Also, the term 'true' ASMR is mentioned for the first time on line 132 but has not been defined in the preceding text. Since it seems to be the case that experiencing 'true' ASMR is used as the main criterion for group assignment, its meaning should first be explained clearly.

Response: 

We have incorporated the definitions of ‘true’ ASMR and false positives in lines 130-133, and 141.

2b: Based on the revised text and the authors responses, it is my understanding that the non-ASMR-experiencer group includes participants who do not report to experience ASMR and also participants who report to experience ASMR but whose experience is not considered 'true' ASMR by the authors. I think that the authors need to explain and motivate their choices for group assignment in the manuscript clearer and in more detail.

Response: 

We have addressed this in our response to section 1c.

Minor issue:

On lines 411-414, it is written that this is the first study to assess whether participants were ASMR-experiencers post-video as well as pre-video. I do not understand what this means. Previous studies have also used questionnaires to 'confirm' or validate participants' ASMR experiencers by asking participants if they consider themselves ASMR experiencers and whether they experienced ASMR while watching a video. How is the method used here different from such studies?

Response:

We agree that many studies use questionnaires to validate ASMR experience, however this technique is typically to categorise ASMR-group allocation prior to the experimental protocol. Furthermore, post-video screening has resulted in participants being excluded if their responses do not match their group allocation (e.g. Fredborg et al., 2018; 2020). In our study, this was a strategy to capture genuine ASMR experiences. ASMR-naïve participants would self-classify as non-experiencers. Many have never heard of or been exposed to ASMR videos, therefore do not have any context by which to evaluate if they are an experiencer or not. In our sample, 8 self-categorised non-experiencers did experience tingles originating in the head and / or neck post-video. Similarly, this technique enabled us to classify 9 true non-experiencers and avoid false-positive reporting. We have identified a paper that has used the same technique thus added this citation (line 139) and removed the wording from lines 411-413.

I still think that the manuscript is interesting and publishing it has value. If group assignment and 'true' ASMR is properly defined and explained by the authors, and further limitations are discussed, I would recommend that it be accepted for publication.

Response:

We would like to thank Reviewer 2 for their thorough comments.

---

## [Decision Letter · Decision Letter 2]

3 Jan 2022

Untangling the tingle: Investigating the association between the Autonomous Sensory Meridian Response (ASMR), neuroticism, and trait & state anxiety

PONE-D-21-21005R2

Dear Dr. Greer,

We’re pleased to inform you that your manuscript has been judged scientifically suitable for publication and will be formally accepted for publication once it meets all outstanding technical requirements.

Kind regards,

Nobuyuki Sakai, Ph.D.

Academic Editor

PLOS ONE

Additional Editor Comments (optional):

Reviewers' comments:

Reviewer's Responses to Questions

**Comments to the Author**

1. If the authors have adequately addressed your comments raised in a previous round of review and you feel that this manuscript is now acceptable for publication, you may indicate that here to bypass the “Comments to the Author” section, enter your conflict of interest statement in the “Confidential to Editor” section, and submit your "Accept" recommendation.

Reviewer #2: All comments have been addressed

2. Is the manuscript technically sound, and do the data support the conclusions?

Reviewer #2: Yes

3. Has the statistical analysis been performed appropriately and rigorously? 

Reviewer #2: I Don't Know

4. Have the authors made all data underlying the findings in their manuscript fully available?

Reviewer #2: Yes

5. Is the manuscript presented in an intelligible fashion and written in standard English?

Reviewer #2: Yes

6. Review Comments to the Author

Reviewer #2: I thank the authors again for taking time to submit a revised version of their paper. I think that the method for differentiating ASMR experiencers from non-experiencers is now clear enough from the manuscript. I find the revisions made based on my previous comments sufficient.

Minor note:

I think that the citation to reference list nr. 37 on line 139 and 414 in the revised manuscript is not clear enough. To my understanding the citation has been included to show that others have used a similar strategy to identify true ASMR experiencers and false-positives (i.e. pre- and post-video/audio reporting), but a naive reader who does not have this information has no way of knowing what the purpose of the citation is. I think it would be important to make the connection between the text and the reference more obvious.

7. PLOS authors have the option to publish the peer review history of their article (what does this mean?). If published, this will include your full peer review and any attached files.

Reviewer #2: No

---

## [Editor Report · Acceptance letter]

7 Jan 2022

PONE-D-21-21005R2 

Untangling the tingle: Investigating the association between the Autonomous Sensory Meridian Response (ASMR), neuroticism, and trait & state anxiety 

Dear Dr. Greer:

I'm pleased to inform you that your manuscript has been deemed suitable for publication in PLOS ONE. Congratulations! Your manuscript is now with our production department. 

Kind regards, 

on behalf of

Dr. Nobuyuki Sakai 

Academic Editor

PLOS ONE